# *Mental health is just an Addendum*: Assessing stakeholder's perceptions on COVID-19 and mental health services provision in Malawi

**Martina Mchenga**[1]*, **Yamikani Ndasauka**[2], **Fiskani Kondowe**[3], **Jimmy Kainja**[4], **Chilungamo M'manga**[5], **Limbika Maliwichi**[6], **Simunye Nyamali**[7]

1 Centre for Social Science Research (CSSR), University of Cape Town, Cape Town, South Africa,
2 Department of Philosophy, University of Malawi, Zomba, Malawi, 3 Department of Mathematics, University of Malawi, Zomba, Malawi, 4 Department of Media and Communication Studies, Chancellor College, University of Malawi, Zomba, Malawi, 5 Department of Behavioural and Human Biological Sciences, Kamuzu University of Health Sciences, Lilongwe, Malawi, 6 Department of Psychology and Medical Humanities, University of Malawi, Zomba, Malawi, 7 DoJob, Lilongwe, Malawi

* martinamchenga@gmail.com

**Data Availability Statement:** The study is a qualitative study and interviews were recorded. All relevant data are therefore within the manuscript.

## Abstract

### Introduction

The World Health Organization declared COVID-19 as a pandemic in March 2020. COVID-19 has since caused a significant increase in mental health problems at national and global levels. This study assessed the views of key mental health stakeholders regarding the state of mental health service provision in Malawi and the pandemic's impact on the sector.

### Methods

The study utilised a qualitative approach through key informant interviews (KIIs) conducted using a semi-structured interview guide. The interviews were audio recorded in English language and were manually transcribed for thematic analysis by generating codes re-classified into themes, sub-themes and quotes.

### Results

The results are categorised into five themes. Firstly, the availability of mental health services. All experts confirmed the lack of availability of the mental health services especially at the lower levels of care. Currently, only 0.3% of facilities offer mental health services in Malawi. Moreover, although mental health services are part of the essential health care package and, therefore, are supposed to be provided for free in public facilities at all levels, the services are centralised and only functional at a tertiary level of care in public facilities. Secondly, funding sources for mental health in public and private facilities. We learnt that public facilities depend on donor sources and there is lack of prioritisation in budget allocation for mental health services. Whereas private facilities, their major source of funding is user fees. Thirdly, government's response in the provision of mental health services during COVID-19. Almost all experts echoed that government took a proactive approach to address the mental health needs of its population during the pandemic. There

What this means is that, the transcribed data makes the results section. There are no separate files to be uploaded as all data were discarded after the transcription process ended to ensure anonymity. The respondents provided consent for the transcribed data to be published in the form of results and we assured them that the recordings will be discarded after, which we did. The restrictions are in the sense that the specific details of the respondents beyond those shared in the manuscript can not provided. Otherwise the manuscript provides all the data transcribed from the recordings in the results section.

**Funding:** This study was part of the project funded by the National Research Fund (NRF) of South Africa, grant number COV19200603527586. The funders had no role in study design, data collection and analysis, decision to publish, or preparation of the manuscript

**Competing interests:** I have read the journal's policy and the authors of this manuscript have no competing interests

was increased collaboration between the government and the private sector to provide psychosocial and counselling services to health workers working directly with COVID-19 patients in isolation centres. Furthermore, to increase awareness of the general population on where to seek counselling services. Lastly, challenges in the provision of mental health services were highlighted and how the pandemic acerbated the challenges including shortage in human resources for health and inadequate funding.

## Conclusion

This study underscores the urgency of addressing mental health challenges in Malawi. Policymakers must prioritize the decentralization of mental health services, explore funding opportunities, and build on the successful collaboration with the private sector. These measures will not only enhance the accessibility and quality of mental health services but also ensure that mental well-being is a central component of public health efforts in Malawi.

## Introduction

Globally, it is estimated that one in four people develops a mental disorder at some point in their lives [1]. The most common mental health disorders include anxiety, depressive symptoms, and unexplained somatic symptoms [2], these account for 14% of the total disease burden [3]. In 2019, a study conducted by the Global Burden of Disease (GBD) showed that the global number of Disability-adjusted life years (DALYs) due to mental disorders increased to 125.3 million from 80.8 million in 1990, placing mental disorders among the top ten leading causes of burden worldwide, with no evidence of a global reduction in the burden since 1990 [4]. The greatest burden of mental health disorders (over 80%), however, resides in low- and middle-income countries (LMICs) [4, 5]. In Malawi, it is estimated that mental health disorders account for almost a quarter of the total burden of ill health [6]. Specifically, depression is a widespread mental disorder in the general Malawi population, ranging between 20–30% [6], whereas, among young people, the rates vary between 10–20% [6, 7]. This makes mental health a clear issue for development in low- and middle-income countries requiring urgent interventions [8].

COVID-19 was declared a pandemic by the World Health Organization (WHO) in March 2020, and since then, there has been a dramatic rise in the prevalence of mental health problems both at national and global levels [9]. In Malawi, for example, since the pandemic started, the country recorded an increase in suicide rates by up to 57% [10]. One of the channels through which COVID-19 has been shown to exacerbate mental health disorders is the measures adopted globally to contain the pandemic including national lockdowns, social distancing, and quarantine [9]. In Malawi, Psychologists associate the increase in suicidal rates to be due to the loss of jobs and rising poverty levels during the pandemic which put intense pressure on family welfare [10].

Mental health disorders are included in the United Nations Sustainable Development Goals (SDGs), calling out all member states to consider mental health as a priority for global development [11]. Malawi is a signatory to the UN SDGs and is therefore committed to ensuring quality health for all including the provision of quality and affordable mental health services [12]. Thus, mental health services in Malawi are provided as part of the essential health care package (EHP) and are accessible in public and selected Christian Association of Malawi (CHAM)

facilitiesfree at the point of use. However, EHP services in Malawi are inadequate due to a lack of funding among other challenges. The existing evidence shows that COVID-19 exacerbated existing health systems' challenges thereby affecting effective delivery of essential service, especially in low resource settings [13]. This paper, therefore, sought to understand from the stakeholders' perspectives the state of mental health services provision in a context where resources are limited and how the COVID-19 pandemic affected the provision of the services. We hope that through this study we are engaging with the dialogue on the importance of investing in mental health services in LMICs and strengthen the system for effective provision of the services in the wake of the pandemic and beyond.

## Methodology

### Study setting

The health system in Malawi is centrally controlled by the Ministry of Health. Zonal offices provide technical support to district health offices, which, in turn, oversee healthcare operations in all 28 districts. Nearly all of these districts (24 out of 28 districts) have their own district hospitals [12]. Within each district, health centres, are predominantly staffed by paramedics and nursing personnel, serving as referral centers for health posts managed by health surveillance assistants (HSAs). Most districts, have at least one medical doctor, who usually performs administrative duties as a district health officer [12]. There is a central hospital in each of the three regions; these act as tertiary units for the district hospitals in their regions [12]. The directors of these central hospitals report directly to the Ministry of Health.

Mental health services in Malawi trace their origins back to 1910, initially under the jurisdiction of the prison services [14]. However, in 1951, responsibility for these services shifted to the Ministry of Health [14]. In 1953, a national psychiatric hospital was established in the southern region, located in Zomba, equipped with approximately 333 beds and admitting between 1,500 to 2,000 patients annually [14]. The central region was initially served by Bwaila Hospital's psychiatric unit (under the authority of Kamuzu Central Hospital), a 25-bed facility, and another 50-bed facility run by St. John of God Community Services, both of which are in the country's capital, Lilongwe. However, in 2017, Bwaila Mental Hospital was closed because of poor conditions and a severe shortage of personnel. Zomba is currently the only public health facility and only facility that handles long-term care, although public care is provided at St John of God both in the central and northern region through an agreement with the government of Malawi. Overall, only 0.3% of the facilities both public and private provide mental health services in Malawi [14].

### Study design

The study applied qualitative methods approach allowing us to capture stakeholder's views in a way that we could not quantitatively, and provided us a starting point for future quantitative analysis where feasible. We adopted a two-step approach. First, the lead author MM conducted rapid desk review of relevant literature to identify the current topics on mental health services provision in low and middle income settings and the impact of the COVID-19 pandemic on the sector. This part informed the development of the semi-structured interview guide. Second, using the interview guide, we conducted in-depth key informant interviews with major stakeholders from government and the private sector who have expert knowledge of the Malawi health system and are involved in the provision of mental health services in Malawi.

The study adopted pragmatism as an interpretative framework by placing the importance on the problem that we were studying, the questions asked and the practical implications of our research [15, 16]. Based on this framework, the study was also guided by the ontological

belief that reality is what is useful and practical, and the epistemological belief that reality is known using multiple objective and subjective approaches [15, 16].

We conducted a formative qualitative assessment using key informant interviews (KIIs) among stakeholders with expert knowledge and involvement of mental health services in Malawi. Data were collected till saturation point. We chose key informant interviews because unlike focused group discussions we wanted to get more candid and in-depth answers even on sensitive issues. All authors who have a minimum qualification of a master's level training conducted the interviews. Two of the co-authors CM and LM have primary training in clinical psychology and were also instrumental in the development of the semi-structured topic guides which facilitated the discussions.

## Study population and sampling

Participants were recruited using purposive sampling and below is the inclusion criteria applied:

1. Providers working in private not profit facilities which provide mental health services.

2. Psychiatric nurses working in both government and private health facilities

3. District health officers in facilities where mental health services are provided.

4. Director at the Ministry of health.

5. Private for profit mental health services providers.

6. Director of mental health services at the Zomba mental hospital.

We started out by sending official request letters through emails to the potential participants to introduce the project and the team. This was then later followed by a telephone call. Individual respondents who expressed interest in participating were requested to provide the date and time which would work best for them to conduct in person interviews. Before the interviews, the participants were provided with a consent form. Data on institution affiliation, gender and position were collected at the beginning of the interview.

Nine stakeholders were interviewed for the survey, a number that aligns with the relatively limited scale of the mental health sector in Malawi. Crucially, we reached a saturation point at this sample size, as common themes consistently emerged during the interviews. Saturation pertains to the point at which further data collection becomes 'counter-productive,' offering diminishing returns and failing to significantly contribute to the overarching narrative or theory [17]. We viewed the number nine as the saturation point where additional data collection yields 'diminishing returns' [18].

Furthermore, a systematic review of qualitative research and saturation points found the range to typically fall between 9 and 17 [18]. Given the specific nature of our study and the relatively small size of our study population, we are confident that we attained our saturation point at an appropriate number. Although our sample size was modest, it effectively represented key mental health stakeholders from both the public and private sectors, offering diverse perspectives and enriching our policy recommendations.

## Data collection process and tools

Data collection was conducted from 12th May-30th June, 2021. The interview guide was developed based on key issues highlighted in the literature and inputs of psychologists from academia and psychiatric medical personnel from the Ministry of Health with knowledge and expertise in mental health. The guide was reviewed by authors CM and LM who have practical

experience and expertise in the mental health field for content and construct validity. Furthermore, the questionnaire was piloted to test the adequacy, relevancy and to ensure that data generated reflected the research aim.

The KII semi-structured interview assessed the respondent's awareness of mental health guidelines, policies, and strategies in Malawi, the distribution, availability and access to mental health services, availability of resources for mental health services i.e., human and financial resources, their organisation's role in the provision of mental health services, the impact of COVID-19 on the access and provision of mental health service delivery, and what are the gaps and opportunities experienced in the provision of mental health services resulted from COVID-19 pandemic.

In essence the interview guide was developed and used to safeguard consistency [19] based on the research questions and review of the literature. It helped ensure that our interviews addressed themes identified in advance in the literature as crucial to the research questions [20, 21]. As a result of the nascent nature of health systems in Malawi and other LMICs alike, we adopted open-ended semi-structured interviews. Unlike forced choices in surveys, open-ended inquiry enhances the accuracy of retrospective reports because participants are at liberty to state whether they remember certain events or not [21]. Furthermore, semi-structured interview allows research participants to give a vivid description of their feelings, experience and views about the importance of a topic- in this case the state of mental health services in Malawi.

To minimise bias from both the interviewer and interviewee, we adopted the funnel approach of interviewing, building of trust, rapport and presenting questions in an unbiased manner [22]. Interviews were conducted in English. The interviews were approximately 1.5 hours and were conducted face-to-face at a location and time of the participant's preference. No non-participants were present at the interviews. Field notes were made after the interview regarding the general impression of the interviews. Transcripts were returned to the participants for comments and data validation.

## Data analysis

Authors of this paper analysed the stakeholders interviews using both an inductive and deductive approach [23]. Before the transcription, all personal identifiers were removed from the transcripts. The audio recordings were first transcribed manually to MS Word by a commercial transcription services provider. MM and YN repeatedly listened to the audio recordings to check the accuracy of the transcription and reread the transcripts line-by-line to capture emerging themes (inductive approach). The authors also used the main research questions to help guide the textual analysis (deductive approach). Themes were then organized into categories, followed by codes and sub-codes. To ensure consistency during data analysis, MM and YN independently developed an initial coding framework based on the questions used to facilitate the discussions, the two authors then compared and harmonized their coding frameworks and used insights from the anonymised transcribed data to refine and modify the coding framework. In addition, two independent coders performed an intercoder agreement exercise before the final codebook was generated [24]. In addition to developing a codebook, the coders analysed commonalities and differences by themes, as well as the intersections between themes.

## Ethics statement

Ethical clearance for the study was granted by the University of Malawi Research ethics committee (UNIMAREC) (No. P/03/21/53). The background, procedure, and aims of the study were communicated orally to all respondents, along with an assurance that information would

be kept confidential and that no payment would be given for participating. Respondents were not identified by name in any transcript, report, or publication in order to maintain anonymity. Written informed consent was sought from all participants. The signing of consent was performed after the participant had read the study information sheet.

## Results

In this section, we present study findings starting with description of the study population and sample size.

### Population characteristics

Our sample included representatives from national government responsible for defining national health policy in the country and based at the Ministry of Health headquarters; private not-for-profit stakeholders who work in collaboration with the government in the provision of mental health services; private for profit mental health providers who work independently and own their practice; district health officers who manage government facilities at the district level. The majority of the participants were female (Table 1).

### Main findings

The main findings are organized in four thematic areas namely availability of mental health services; funding sources of mental health services; impact of COVID-19 on mental health services provision and general challenges. The main themes and subthemes that came out from the interviews are presented in Table 2.

### Distribution, availability, and access to mental health services

In this section, we delve into a comprehensive stakeholder analysis of the distribution, availability, and accessibility of mental health services in Malawi.

**Availability and distribution of mental health services in Malawi.** Ideally, mental health services are supposed to be provided at all levels of care, from primary level to tertiary level in all public facilities. However, all stakeholders confirmed that, at the time of the survey, mental health services were accessible at the district level of care within only one district facility in Mzuzu (northern region) and two tertiary facilities (Zomba mental hospital in the southern region and ST John of Hope in Lilongwe, central region). Zomba mental hospital is publicly funded whereas the other two are mission/ private not for profit. The distribution of these services was a topic of concern, as all stakeholders voiced unanimous concern about the highly centralized and unequal distribution of mental health services. Notably, all three facilities

**Table 1. Description of the policymakers interviewed.**

| No | Position | Gender | Institution affiliation |
|---|---|---|---|
| KI1 | Programme manager | Female | Private not for profit facility |
| KI2 | Psychiatric nurse | Female | Government facility |
| KI3 | Psychiatric nurse | Female | Government facility |
| KI4 | Director | Male | Ministry of Health |
| KI5 | District Health and Social Service Officer | Male | District hospital |
| KI6 | Private Mental Health Counsellor/Psychiatric consultant | Male | Private practice |
| KI7 | District Health Officer | Female | Ministry of Health |
| KI8 | Psychiatric nurse | Male | Mental health association of Malawi |
| KI9 | Mental hospital director | Female | Government tertiary hospital |

**Table 2. Themes and subthemes from the key informant interview with stakeholders in the mental health sector.**

| Themes | Distribution, availability, and access to mental health services | Awareness of mental health guidelines, policies, and delivery strategies | Availability of resources | Impact of COVID-19 on provision and utilisation of mental health services | Gaps and opportunities |
|---|---|---|---|---|---|
| Subthemes | • Availability and distribution of mental health services<br>• Accessibility of mental health services | • Mental health policies, guidelines, and national laws<br>• Delivery strategies and referral system | • Financial resources<br>• Human resources for health | • Government of Malawi response to COVID-19 in relation to Mental health<br>• Impact of COVID-19 on mental health services utilization | • Stakeholder collaboration<br>• Investment in human resources for health in the mental health sector |

providing mental health services at a higher level are located in urban areas. Furthermore, private for-profit service providers mainly operate at a very small scale and predominantly focused on urban populations, targeting those with the financial means to afford such services as they require user fees at the point of service.

At primary level of care, the focus is to provide the health workers with first aid training on how to handle clients with serious psychotic symptoms. This was explained by one KI below:

*At the primary level, mental health illness assessment skills are taken as first aid; that's why there is no specialized care at that level. The expectation for every health worker at this level, whether a nurse, clinical officer, or medical assistant, is the ability to assess and provide treatment to calm a psychotic/aggressive patient before referring them to a higher facility level (KI 2).*

**Accessibility of mental health services in Malawi.** In unanimous agreement, all stakeholders acknowledged the limited accessibility of mental health services in Malawi, particularly due to their absence in public facilities and their predominant concentration in urban areas. However, in an effort to improve access to inpatient mental health care, the government forged an agreement with the St. John of Hope. This agreement involved the removal of user fees for selected mental health services, as described by a key informant from St. John of Hope:

*"Following the closure of the Bwaila mental health service section in Lilongwe, the government approached us to establish a service level agreement. Under this agreement, the government committed to providing financial support for staff salaries in exchange for the elimination of user fees. Consequently, we have designated a specific number of beds for patients referred by the government, and these services are provided free of charge."*

## Awareness of mental health guidelines, policies, and delivery strategies in Malawi

**Awareness of mental health guidelines and national laws.** Mental health guidelines and national laws play a pivotal role in shaping the delivery of healthcare services, offering indispensable guidance on how to provide these services efficiently and effectively. Our respondents affirmed the existence of several key documents that serve as guiding lights in the realm of mental health services in Malawi.

1. The Malawi Standard Treatment Act: Historically, the provision of mental health services was bolstered by the Malawi Standard Treatment Act, a document with roots dating back to 1948. This act offered detailed insights into symptoms, signs, and treatment protocols for mental health-related conditions, and it provided overarching guidance for the delivery of

mental health services. At the time of our interview, it was disclosed that efforts were underway to revise this act, ensuring alignment with the contemporary state of healthcare services.

2. The Mental Health Handbook: Developed in collaboration with the Scotland Malawi Health Education Partnership, the Mental Health Handbook serves as a cornerstone reference for practitioners. This handbook, specifically designed to cater to general practitioners without prior mental health training, is readily available across various healthcare facilities at all levels. In a significant step towards enhancing accessibility, this guide was also made available in digital format and formally launched in 2020.

3. Health Sector Strategic Plan (HSSPII): The Health Sector Strategic Plan (HSSPII) further reinforces the critical importance of mental health service provision by specifically addressing mental health illnesses. This strategic plan not only recognizes the significance of mental health but also furnishes valuable guidance on the implementation of interventions aimed at alleviating the burden of mental health.

**Delivery strategies for mental health services and the referral system.**   Mental health services in Malawi are organized within a tiered system of care, intended to provide comprehensive support across primary, secondary, and tertiary levels. However, the current landscape reveals unique challenges and approaches at each level, shaping the delivery of mental health services in the country.

Ideally, mental health services are supposed to be integrated across all three levels of care, however, all our participants indicated the services are non- existent at lower levels of care. As previously mentioned, mental health services in Malawi are identified with specialist or higher level of care, so much that health care providers at primary levels of care in public facilities lack skills to effectively provide mental health services of any kind. The focus at this level is to provide the health workers with first aid training on how to handle clients with serious psychotic symptoms. This was explained by one KI below:

*At the primary level, mental health illness assessment skills are taken as first aid; that's why there is no specialized care at that level. The expectation for every health worker at this level, whether a nurse, clinical officer, or medical assistant, is the ability to assess and provide treatment to calm a psychotic/aggressive patient before referring them to a higher facility level* (KI 2).

At the secondary level of care, not much is done in government facilities, the most of what happens is outpatient visits. Like primary facilities, district hospitals do not have individual or separate units for mental health illnesses where patients can be admitted and managed. However, some district hospitals reserve few beds for inpatient care for the mentally ill. Besides a nurse specializing in mental health at this level of care, there is also at least one psychiatric clinical officer and a mental health program coordinator. The coordinator is responsible for coordinating mental health functions. Although, unlike the primary level, the districts have cadres with training in mental health services provision, nonetheless, due to health care worker shortages, they are also involved in other tasks such as maternal and child health services, which usually affect the level of mental health services quality. The model of treatment is also largely biomedical. In the private not-for-profit sector, there is only one secondary level facility in Mzuzu managed by St John of God. The facility provides inpatient care services to clients with acute mental health illness and provides both biomedical treatment and psycho-social services.

However, it is very limited in capacity and its focus is providing community mental health services and special needs for children with disabilities.

At the tertiary level, specialists in mental illnesses are available, and as such specialized mental health services are provided. There are two tertiary facilities in Malawi. Zomba Mental Hospital, located in the Southern Region and mainly targets clients from the Southern Region and is managed by the government. Lilongwe St. John God of Hope, a private not-for-profit facility operating under the Christian Association of Malawi (CHAM) which carters to clients mainly from the central region. The type of specialists available at the tertiary facilities, however, differs by the provider. As previously mentioned, government mainly focuses on severe forms of mental illnesses that manifest physically and so the type of health workers are mainly Psychiatrist with a focus on biomedical model of treatment. On the other hand, St John of God provide services for all forms of mental health illness and uses a bio-psychosocial model providing counselling services to clients with psychosocial mental illnesses.

Besides the hospital-based mental health services, outreach clinics run by government facilities and St John of God are available to bring community awareness around mental health related illnesses. For instance, Zomba Mental Health hospital runs some outreach clinics. Lilongwe District Health Office (DHO) also facilitates some outreach programs, although it is not yet documented to what level this is done. On the other hand, St John of God also operates seven outreach clinics yearly in the Central and Northern Regions.

To fill in the gaps in mental health services provision, the Ministry of Health organises community outreach programs to bring awareness of the service available at the district facility. These outreach programmes are conducted by health workers from the district facilities (secondary level of care) who have had some training in mental health. Ideally, the outreach programmes are supposed to be conducted monthly, however due to lack of funding this does not always happen. Again, even at this level, the focus is on bringing awareness around psychotic and severe cases of mental illnesses.

Like any other healthcare service, mental health services are linked through an elaborate referral system where patients are referred from the primary health facility to the district facility, from which they get directed to the tertiary facility. Referral to St John of God facilities is made through a government facility and allows clients referred to access the services free of charge through the SLAs, otherwise clients are expected to pay. Ideally, the way the referral would work is for clients in the northern region, their referral hospital would be St John of God in Mzuzu, those from the southern region, Zomba Mental and central region would be the St John of Hope Hospital in Lilongwe, However, as one key respondent said, the referral system does not always work in this ideal way because of; (i) lack of an effective gatekeeping system and (ii) non-existence of the services at the lower level which affects efficient service provision especially in the public facilities.

*I am also aware that some clients are referred to Zomba Mental Hospital from the North and others from the Central Region, and these are clients that mainly require admission because sometimes St John of God in these two Regions does not have space for admission. But ideally, they are supposed to go to facilities within their respective Regions (KI 1).*

## Availability of resources for mental health services

### Funding for mental health services in Malawi

The WHO has previously highlighted the chronic underfunding of mental health in countries such as Malawi [9]. Thus, before the COVID-19 pandemic, countries were already spending less than 2% of their national health budgets on mental health and struggling to meet their populations' needs [9]. Under this theme, stakeholders revealed the sources of funding for mental health services by different providers and discussed the challenges they face to get adequate funding.

**Mental health funding at public facilities.**   Most of the key respondents highlighted that funding for mental health services in public facilities is inadequate and grossly insufficient as the health sector is majorly donor dependent. Currently, mental health services are not prioritized and are thus funded through other recurrent transaction (ORT) expenditures, which is concerning as it is not reliable and sustainable. As noted by one respondent, the problem with the current arrangement is that mental health services do not have a particular department/unit and fall under non-communicable diseases. As a result, there is no specific budget allocated to mental health services, and usually, the funds allocated are inadequate. This affects facilities in the districts the most and mental health programs are also affected.

> *In the case of us in the districts, funds available for mental health services are less than USD100 monthly. . . the funds are not enough, so we don't always conduct the outreach clinics. However, we wished to have more outreaches to support the hard-to-reach areas (KI 3).*

**Funding for mental health services at private not for profit facilities.**   Unlike public providers, the two St John of God facilities have multiple sources of funds. Among them include the government through the service level agreement (SLA). Under the SLA, the government pays for residential healthcare for patients referred from a public facility to St John of God. The government also supports St John of God with staff salaries. However, the current SLA agreement has its challenges as highlighted:

> *In the current SLAs arrangement, the government only pays for accommodation for the referred patients and not their medication. However, mental health drugs are expensive. For example, the minimum cost per patient, including medicine, is about USD750, whereas the SLAs only cover about USD250, which is the standard cost (KI 2).*

The other means of funding include grants through proposals to different funders for programs such as mental health promotion and rehabilitation; local fundraising by renting out conference facilities; donors/proprietors who conduct several fundraising activities; and client user fees.

**Human resources for health.**   Inadequate human resources for health is a persistent challenge in Malawi especially in the public facilities [25]. With the coming of the pandemic, it was evident how lacking the facilities are to effectively provide mental health services at a time when the people needed them the most. As highlighted, mental health services at lower levels of care are non-existent, and even though services are available at the secondary level, health workers do not have the necessary skills and support to effectively provide the services. A district hospital officer managing one of the public facilities attested to this.

> *Staffing is inadequate, there are only four providers here, so this number is minimal to cater to patients. In addition, we do not manage to do other services in this district. We are primarily*

*occupied in the wards, so we agree that per month each person should cover each department at least once a week: e.g., mental health, and other weeks covering the other wards. And we only have one counsellor, one certified psychosocial counsellor, in the whole district. So, she is usually not there in the community; she is always at the district facility (KI8).*

Another existing challenge regarding health workers in the mental health field is that very few health workers are registered with the Medical Council of Malawi to provide psychosocial services and other treatments. This situation contributes to the shortage of health workers in the general and mental health fields.

## The impact of COVID-19 on the access and provision of mental health service delivery during pandemic

Anecdote evidence shows that the pandemic increased the demand for mental health services across the global. The same was reported true in Malawi. Although we could not get the actual figures for the increase, one of the common sentiments among the respondents was that, in general, despite increased cases of depression and anxiety during the pandemic, utilization rates for the services have not changed much. One possible reason given was that the measures put in place to control the pandemic such as isolation and social distance, made it difficult for people to access the services.

*Due to measures put in place to control the COVID-19 pandemic, there was a disruption of some essential services at the district and central levels. Even the availability of essential medicines was interrupted due to logistical challenges. In addition, the availability of health workers at the community level was disrupted and some conceptions of health workers visiting the communities people feared that they were bringing in COVID-19 (KI 9).*

The government of Malawi took several initiatives in response to the psychological impact that COVID-19 had on the general population as well as health workers. We were told that the government of Malawi proactively stepped up to make mental health services available and accessible by collaborating with St John of God to provide psychosocial and counselling services to health workers working directly with COVID-19 patients in isolation centres. This is articulated by KI4 below.

*The government made health services available for health workers working at isolation centres after noticing how overwhelmed and mentally exhausted the health workers were. This was done with the help of St. John of God (KI 4).*

For the general population, the government also increased awareness of other forms of mental illnesses besides those that manifest physically through different media outlets and provided list of service providers where people can access psychosocial services. Just like health workers, the public could also access counselling services from St John of Hope as well as a list of private practitioners which the government collaborated at an affordable cost.

The private not for profit sector also got involved by being part of the COVID-19 task force preparedness committee and took part in awareness campaigns using their resources to disseminate messages on COVID-19 and how people have been affected by the pandemic, and where to get help. As KII 2 noted:

*We have also been facilitating mental and psychosocial first aid training for practitioners. Apart from the general mental health services that we are providing, we have also seen that*

*there are a lot of mental health issues due to COVID-19, such as stress, depression, and anxiety. We have a stress management clinic that offers psychosocial counselling support. We have utilized more of that to provide support to people who have been coming in with COVID-19-related mental issues. We also provide OPD and in-patient support, and we have seen an increase in the number of people, although we can't tell the specific numbers (KI 2).*

To eliminate the access barrier that COVID-19 response measures created in accessing mental health services, St John of God for example, adopted the use of technology such as mobile phone consultations and teleconferencing. However, these types of technologies are only accessible to the select few in Malawi. The majority especially the rural poor cannot afford them as they require mobile data and internet.

Looking at the longer term and realising the need for more mental health workers in the public facilities, the government took extra steps to advertise for new positions with a focus on recruiting health workers with expertise in psychosocial counselling services. It is however, yet to be seen what impact the changes made will have on the efficiency and quality of mental healthcare services in public facilities.

## The gaps and opportunities experienced in the provision of mental health services resulted from COVID-19 pandemic

Regarding opportunities, respondents shared the sentiments that COVID-19, in a positive way, has helped put together various stakeholders in the districts to work in a collaborative way. For example, several partners are now working together in planning and resource mobilization for mental health illnesses. It also created opportunities for increased partnership between the government and the private sector by contracting them to provide psychosocial services to health workers and the public.

Further, the pandemic has also brought to light how important investing in mental health is and that there is a need for providers to put systems in place that will ensure effective provision of the services even in the face of the pandemic. For example, St John of hope adopted new technologies to ensure continuity in mental health services provision as articulated by KI2 below.

*COVID-19 has forced mental health service providers to be more innovative. For example, the use of virtual meetings to reach out to people without meeting them physically. It has also stimulated my interest to research the mental health area and develop innovative interventions. Besides government now has advertised positions and postings specific to mental health, which were not there before (KI2).*

The government on the other hand, in response to the shortages in health workers with mental health related specialisation, advertised new positions specific for mental health related services to be deployed in different districts.

## Discussion

In this paper, we conducted interviews with relevant stakeholders in the mental health sector to assess their perceptions on the state of mental health services provision and how the pandemic affected the provision of the services in Malawi. Key issues related to availability and distribution of mental health services, awareness of mental health guidelines, policies and national levels, availability of resources, impact of COVID-19 on the utilisation and provision of mental health services as well as opportunities and gaps were highlighted.

First, we found that although mental health services are part of the essential health care package [6] and, ideally supposed to be accessible at lower levels of care including primary care and rural areas, there are non-existent. This observation aligns with Udedi's argument [6], asserting that in sub-Saharan Africa, despite the majority of the populace residing in rural regions, mental health services are predominantly concentrated in urban areas. Lack of integration of mental health services in all levels of care significantly limits accessibility, particularly for rural and economically disadvantaged populations. In the context of Malawi, this translates to 80% of the population not able to access mental health services when they need to, thereby undermining universal health coverage efforts.

Second, stakeholder also highlighted the lack of adequate skills and specialist health workers in the mental health space. Challenges in human resources in the health sector has been a long-standing issue in Malawi and when it comes to mental health, the situation is even worse. Currently, there are 0.01 psychiatrists per 100,000 people [6], figures much lower comparable to other countries in the sub-Saharan Africa region including those of Ghana (0.058), Rwanda (0.060), Zambia (0.056), Zimbabwe (0.095) [26]. Furthermore, most general healthcare workers lack the competence and confidence to effectively handle psychiatric patients. The COVID-19 pandemic has further exposed the inadequacies in the country's healthcare workforce, prompting the Malawian government to initiate recruitment drives for positions in counselling and psychiatry. However, it remains uncertain whether the recruitment process has been completed and whether the deployment of personnel has been equitable, particularly in extending coverage to lower-level facilities.

As noted by all stakeholders, the historical underfunding is a significant contributor to the existing challenges in the provision of mental health services in Malawi. In Malawi, mental health services are categorized under non-communicable diseases (NCDs) and receive funding as a portion allocated to the NCDs department. However, this funding is often insufficient, with a predominant portion directed towards higher-level facilities. Consequently, this underfunding has led to a fragmented mental health system, where mental health is not appropriately prioritized within the broader healthcare framework. This situation results in insufficient resources, limited training for healthcare workers, and a disjointed approach to delivering mental health care.

In certain countries of sub-Saharan Africa, such as South Africa [27] and Ghana [28], effective interventions have been implemented to overcome resource limitations. These interventions include task-shifting strategies, wherein non-specialist healthcare workers are trained to deliver basic mental health services, thus broadening access to care beyond urban areas. To enhance mental health service delivery in Malawi, similar strategies can be adopted. These may include task-shifting, community-based interventions, and collaborative efforts across multiple sectors. By investing in training initiatives, increasing funding allocations, and enhancing the integration of mental health into primary healthcare systems, Malawi can enhance access to mental health services and tackle the prevailing challenges within the sector.

Lastly, although the pandemic put extra pressure on the already limited resources and exposed the gaps that exist, we found that it created opportunities and strengthened the collaboration between the public and the private sector. Given that public facilities are lacking to effectively provide psychosocial services, besides capacity building, partnerships with the private partnership can be effective making mental health services especially counselling and psychosocial accessible to more people. A 2021 scoping review of the role of public private partnerships in the provision of primary care services found that PPPs have the potential to increase access and to facilitate the provision of prevention and treatment services for certain target groups [29]. Moreover, collaborative efforts between governments, non-profit organizations, and international partners have also proven to be effective in addressing mental health

service gaps through community-based interventions, training programs for health workers, and the integration of mental health into primary healthcare services in countries like Uganda [30], South Africa [31] and Kenya [32].

The study had some limitation which warrant mentioning. Firstly, the data used in the analysis is based on self-reports, which could have led to over-reporting and underreporting of some aspects. However, we think that the findings from this study provide a starting point for a dialogue on how best stakeholders can work towards the effective provision of mental health services during and out of pandemics. Secondly, even though we reported on utilization of mental health services, the study would have benefited more from beneficiaries and users of these services.

As such, before any strong recommendations are made, additional studies and analysis are needed in the mental health field to fully unpack the prevalence of mental health disorders in Malawi by sex and age for efficient targeting. We would also need to understand the extent of health system challenges to effectively provide mental health care services, for example, mapping out available mental health providers, skills, and types of services they can provide. This can be done through the inclusion of mental health related questions in the health facility assessment surveys in Malawi. This will quantify the challenge in effective mental health services delivery and provide a clear understanding of the existing gaps in skills which can then inform appropriate interventions. Finally, given the non-existence of mental health services at lower levels of care, it would also be helpful to understand what interventions are being put in place to promote integration of mental health services at all levels. A report by the WHO and WONCA [33] study shows that integrating mental health services into primary care with effective and acceptable interventions increases access to mental health services.

## Conclusion

This qualitative study explored expert stakeholders' perspectives on the state of mental health services in Malawi. Through key informant interviews, we gained insights into the availability, distribution, funding, human resources, and impact of COVID-19 on mental healthcare delivery. Our findings reveal several key issues. First, mental health services are highly centralised and unequally distributed, with limited integration into primary care. Second, chronic underfunding of the mental health sector persists, especially within public facilities. Third, there is a severe shortage of skilled mental health workers across all levels of care. Fourth, COVID-19 exposed and exacerbated these long-standing deficiencies within the mental healthcare system. However, the pandemic also catalysed new opportunities, including strengthened government-private sector partnerships to expand service delivery. Moving forward, decentralising mental health services, securing sustainable financing, building human resource capacity, and leveraging public-private collaboration will be instrumental to enhancing accessibility and quality of care.

## Supporting information

**S1 File.**
(DOCX)

## Author Contributions

**Conceptualization:** Martina Mchenga, Yamikani Ndasauka, Fiskani Kondowe, Jimmy Kainja, Chilungamo M'manga, Limbika Maliwichi, Simunye Nyamali.

**Data curation:** Martina Mchenga, Yamikani Ndasauka, Jimmy Kainja, Chilungamo M'manga.

**Formal analysis:** Martina Mchenga, Fiskani Kondowe.

**Funding acquisition:** Martina Mchenga, Yamikani Ndasauka, Fiskani Kondowe, Jimmy Kainja, Chilungamo M'manga, Limbika Maliwichi, Simunye Nyamali.

**Investigation:** Yamikani Ndasauka, Chilungamo M'manga, Limbika Maliwichi.

**Methodology:** Martina Mchenga, Yamikani Ndasauka, Fiskani Kondowe, Jimmy Kainja.

**Project administration:** Simunye Nyamali.

**Resources:** Martina Mchenga.

**Software:** Martina Mchenga.

**Supervision:** Martina Mchenga, Yamikani Ndasauka, Jimmy Kainja, Limbika Maliwichi, Simunye Nyamali.

**Validation:** Martina Mchenga, Yamikani Ndasauka, Fiskani Kondowe, Jimmy Kainja, Chilungamo M'manga, Limbika Maliwichi, Simunye Nyamali.

**Visualization:** Martina Mchenga, Chilungamo M'manga, Limbika Maliwichi, Simunye Nyamali.

**Writing – original draft:** Martina Mchenga.

**Writing – review & editing:** Martina Mchenga, Yamikani Ndasauka, Fiskani Kondowe, Jimmy Kainja.

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
