## [Decision Letter · Decision Letter 0]

2 Oct 2022

PONE-D-22-22739Mental Health is Just an Addendum: Assessing stakeholders’ perceptions on mental health services provision in Malawi prior to and during COVID-19PLOS ONE

Dear Dr. Mchenga,

Thank you for submitting your manuscript to PLOS ONE. After careful consideration, we feel that it has merit but does not fully meet PLOS ONE’s publication criteria as it currently stands. Therefore, we invite you to submit a revised version of the manuscript that addresses the points raised during the review process.

We look forward to receiving your revised manuscript.

Kind regards,

Aboubacar Sidiki Magassouba, MD

Academic Editor

PLOS ONE

Journal Requirements:

"I have read the journal's policy and the authors of this manuscript have no competing interests"

4. Please amend your manuscript to include your abstract after the title page.

Reviewers' comments:

Reviewer's Responses to Questions

**Comments to the Author**

1. Is the manuscript technically sound, and do the data support the conclusions?

Reviewer #1: Partly

Reviewer #2: Partly

2. Has the statistical analysis been performed appropriately and rigorously? 

Reviewer #1: Yes

Reviewer #2: N/A

3. Have the authors made all data underlying the findings in their manuscript fully available?

Reviewer #1: Yes

Reviewer #2: Yes

4. Is the manuscript presented in an intelligible fashion and written in standard English?

Reviewer #1: Yes

Reviewer #2: Yes

5. Review Comments to the Author

Reviewer #1: The authors address an interesting issue. The results of such an analysis can be used to guide policy in providing mental health services. However, I have concerns about the methodology adopted.

Minor:

The line numbering is missing, which makes it difficult to comment on each part.

Major:

- The authors talk about two approaches to meet the aim of the study; however, only one method is detailed in this article. What about the other method? Are the results available? If so, please present them in the manuscript.

- Although the sample size does not matter in the qualitative study, I wonder if nine people can meet the assigned objective. What is the argument for having only this number?

- The authors discuss the provision of mental health services before and during COVID-19, yet these two periods are not compared in the manuscript. Is there a logic behind this position?

Reviewer #2: This paper describes the lack of mental health services available within the Malawi setting. The title suggests that the paper will be about the effectiveness of mental health services and this is emphasised within both the introduction and conclusions to the paper. Whilst the phrase 'mental health is just an addendum' highlights rightly the lack of funding and prioritisation that has been attributed to mental health service provision in Malawi and indeed its location within the MoH under NCDs, something that has limited efforts to strengthen this area of health services, the paper itself does not move beyond a description of the limited services available, with a focus on availability, funding, the impact of COVID-19 and challenges. I had expected the paper, rather to consider the effectiveness of services, albeit from the perspective of key stakeholders rather than just their availability.

Specific comments on each section of the paper

1. The introduction is very brief and includes some useful background data.

2. The methods are generally but briefly presented and there is no real detail on the positionality of the researchers, who collected the data etc. The text moves between describing the tool as a questionnaire and as an interview guide.

3. There doesn't appear to have been any attempt to conduct an inductive analysis on the data and as a result the deductive topics are likely to be the themes within the results. Whilst a deductive approach to this type of assessment would be a valid one, it would generally be informed by a conceptual framework but this has not been included or referred to.

4. The results are interesting in reinforcing what is already known about the state of mental health services in many LMIC in SSA and describe in some detail the services available by geographic region and by government/private providers. The national policy context is also described, although this was not based on interview data, The lack of funding available is rightly described as insufficient but the authors go no further than this. I wasn't quite convinced by the suggested orientation of the impact of COVID-19 on services within the paper as this was largely focused on general services, highlighting the fact that COVID-19 exacerbated already existent service gaps. There was some description of government efforts to provide services for exhausted HCW but nothing on the increase need for mental health services within general populations. It was nice to hear that COVID has encouraged collaborative working and partnerships across stakeholders in the district but there was no description of how this has beneficially impacted mental health service provision. Perhaps the data is there but it has not been included within the paper. The findings section does not specify types of services provided at each level in any detail and rarely distinguishes between mental health services to address conditions such as anxiety and depression from those addressing psychosis.

5.The study limitations only highlight the self-reported nature of the interviews with key stakeholders and the conclusions introduce new areas that are not included within the findings such as consideration of why people may or may not uptake mental health services based on findings from elsewhere.

I agree that further research is urgently needed in this area including health systems interventions to promote and embed services at all levels but I am not convinced that this paper contributes greatly to solutions, either within the Malawi context or more broadly.

I suggest that the authors review the transcripts/data and conduct a more nuanced analysis in order to strengthen the paper.

6. PLOS authors have the option to publish the peer review history of their article (what does this mean?). If published, this will include your full peer review and any attached files.

Reviewer #1: **Yes: **Almamy Amara Touré

Reviewer #2: No

---

## [Author Response · Author response to Decision Letter 0]

16 Nov 2022

Aboubacar Sidiki Magassouba, MD

Academic Editor

PLOS ONE

16/12/2022

Dear Doctor Magassouba,

Thank you for inviting us to submit a revised draft of our manuscript entitled, " Mental Health is Just an Addendum: Assessing stakeholders’ perceptions on mental health services provision in Malawi prior to and during COVID-19" to PLOS ONE. We also appreciate the time and effort you and each of the reviewers have dedicated to providing insightful feedback on ways to strengthen our paper. Thus, it is with great pleasure that we resubmit our article for further consideration. We have incorporated changes that reflect the detailed suggestions you have graciously provided. We also hope that our edits and the responses we provide below satisfactorily address all the issues and concerns you and the reviewers have noted.

To facilitate your review of our revisions, the following is a point-by-point response to the questions and comments delivered in your letter dated 3 October, 2022.

1. Editor’s suggestions

• A rebuttal letter that responds to each point raised by the academic editor and reviewer(s). You should upload this letter as a separate file labelled 'Response to Reviewers'.

Response- attached is the rebuttal and response letter

• A marked-up copy of your manuscript that highlights changes made to the original version. You should upload this as a separate file labelled 'Revised Manuscript with Track Changes'.

Response-See attached

• An unmarked version of your revised paper without tracked changes. You should upload this as a separate file labelled 'Manuscript'.

Response-This has been attached

• We note that the grant information you provided in the ‘Funding Information’ and ‘Financial Disclosure’ sections do not match. When you resubmit, please ensure that you provide the correct grant numbers for the awards you received for your study in the ‘Funding Information’ section.

Response-This has been revised. Thank you Editor for taking note of this. The study is part of the project supported by National Research Foundation. Grant number: COV19200603527586. 

• Thank you for stating the following in your Competing Interests section: "I have read the journal's policy and the authors of this manuscript have no competing interests"

Response- Thank you

• Please complete your competing Interests on the online submission form to state any Competing Interests. If you have no competing interests, please state "The authors have declared that no competing interests exist.", as detailed online in our guide for authors your cover letter; we will change the online submission form on your behalf.

Response-Thanks. We have done this

• Please amend your manuscript to include your abstract after the title page.

Response-Thank you Editor. We have made the suggested revision

2. Reviewer 1 Comments

• The authors address an interesting issue. The results of such an analysis can be used to guide policy in providing mental health services. However, I have concerns about the methodology adopted. 

Response-Thank you so much for taking time to review the manuscript and also suggesting ways it can be strengthened. 

• The line numbering is missing, which makes it difficult to comment on each part. The authors talk about two approaches to meet the aim of the study; however, only one method is detailed in this article. What about the other method? Are the results available? If so, please present them in the manuscript.

Response- Thank you so much for this question and need for clarification. We have clarified what we meant by the two step approach on page 5. The first step was done to mainly inform the formulation of the interview guide/questionnaire. 

• Although the sample size does not matter in the qualitative study, I wonder if nine people can meet the assigned objective. What is the argument for having only this number?

Response- Yes totally agree with this observation. Going in we were also worried about the same, however, given how small the mental health sector field is in Malawi (only 0.3% of facilities provide mental health services), the sample we ended up with is still representative and comprised of stakeholders from both the public and private. This gave us a robust picture on the state of mental health services in Malawi. We actually reached saturation point in our interviews where the same themes kept emerging from one interview to another.

• The authors discuss the provision of mental health services before and during COVID-19, yet these two periods are not compared in the manuscript. Is there a logic behind this position?

Response-Thank you for this. To clarify, we wanted to understand what the state of mental health services was before the pandemic and if at all the pandemic has influenced the sector given that it increased mental health related illnesses. We have made revisions and clearly state the study objective to avoid any confusion.

3. Reviewer 2 Comments

This paper describes the lack of mental health services available within the Malawi setting. The title suggests that the paper will be about the effectiveness of mental health services and this is emphasised within both the introduction and conclusions to the paper. Whilst the phrase 'mental health is just an addendum' highlights rightly the lack of funding and prioritisation that has been attributed to mental health service provision in Malawi and indeed its location within the MoH under NCDs, something that has limited efforts to strengthen this area of health services, the paper itself does not move beyond a description of the limited services available, with a focus on availability, funding, the impact of COVID-19 and challenges. I had expected the paper, rather to consider the effectiveness of services, albeit from the perspective of key stakeholders rather than just their availability.

Response- We would like to thank you so much for taking the time to review our paper. Truly your valuable suggestions have strengthened it.

Specific comments

• The introduction is very brief and includes some useful background data.

Response-Thank you so much for this. In the context of Malawi, there are mot many studies that have been done on the subject and it was difficult to get information to describe the context and what evidence exist.

• The methods are generally but briefly presented and there is no real detail on the positionality of the researchers, who collected the data etc. The text moves between describing the tool as a questionnaire and as an interview guide.

Response-We have taken time to review the methodology section describing in detail how the data was collected who did what. We have also described the process of developing the questionnaire as well as interpretation of the results. All this can be found in the methods section.

• There doesn't appear to have been any attempt to conduct an inductive analysis on the data and as a result the deductive topics are likely to be the themes within the results. Whilst a deductive approach to this type of assessment would be a valid one, it would generally be informed by a conceptual framework but this has not been included or referred to.

Response-Thank you for noting this. Indeed we did not include a conceptual framework in our earlier manuscript however, we did adopt pragmatism as an interpretative framework and we were guided by the ontological belief (Gilson 2012; Creswell 2007). All this has now been added to the revised manuscript in the methods section. Also, we must apologise for not being clear, we did use both inductive and deductive approaches. Before developing the questionnaire/guide, we conducted a rapid literature review, this process informed the themes in the questionnaire (deductive approach). After the interview and transcription, we read the transcriptive line by line and even went back to the audio recordings to pick up the themes that were imagining (inductive approach). These two approaches informed the process of summarising our results. 

• The results are interesting in reinforcing what is already known about the state of mental health services in many LMIC in SSA and describe in some detail the services available by geographic region and by government/private providers. The national policy context is also described, although this was not based on interview data, The lack of funding available is rightly described as insufficient but the authors go no further than this. I wasn't quite convinced by the suggested orientation of the impact of COVID-19 on services within the paper as this was largely focused on general services, highlighting the fact that COVID-19 exacerbated already existent service gaps. There was some description of government efforts to provide services for exhausted HCW but nothing on the increase need for mental health services within general populations. It was nice to hear that COVID has encouraged collaborative working and partnerships across stakeholders in the district but there was no description of how this has beneficially impacted mental health service provision. Perhaps the data is there but it has not been included within the paper. The findings section does not specify types of services provided at each level in any detail and rarely distinguishes between mental health services to address conditions such as anxiety and depression from those addressing psychosis.

Response- Thank you so much for this important comment. The type of mental health services provided in Malawi differ by the type of provider. Mental health services in public facilities, are mainly identified with the tertiary hospital which is Zomba mental hospital in the southern region and the model of delivery is through medication. The focus is mainly on severe cases of psychosis, psychotic/ schizophrenic, bipolar illnesses. Because of this even health workers are lower level of care are mainly trained on how to handle patients that have severe sympotms and aggregive. On the other hand, the private not for profit which is mainly St John of God facilities (one in Lilongwe, the central region and the other in Mzuzu in the northern region) besides handling psychotic cases, their model of treatment is both psychosocial and biomedical. We have included this discussion in the results section.

• The study limitations only highlight the self-reported nature of the interviews with key stakeholders and the conclusions introduce new areas that are not included within the findings such as consideration of why people may or may not uptake mental health services based on findings from elsewhere. I agree that further research is urgently needed in this area including health systems interventions to promote and embed services at all levels but I am not convinced that this paper contributes greatly to solutions, either within the Malawi context or more broadly.

Response-Thank you for this. The goal of the paper mainly is to highlight the issues that exist from the actors in the mental health field in Malawi. Our hope for the study was that we would contribute to the dialogue and engagement on the need for LMICs to invest and prioritise more on mental health. In the case of Malawi, there is need to invest in more health workers with psychosocial and counselling skills and go beyond the biomedical model of treatment. And given that COVID-19 encouraged government collaboration with the private sector, the findings reveal that PPPs could be beneficial in improving efficiency and access of mental health services.

• I suggest that the authors review the transcripts/data and conduct a more nuanced analysis in order to strengthen the paper.

Response-Thank you so much for your suggestions and comments which have truly made the paper strong and helped us to go back to the transcripts with a fresher understanding of the concepts.

Again, thank you for giving us the opportunity to strengthen our manuscript with your valuable comments and queries. We have worked hard to incorporate your feedback and hope that these revisions persuade you to accept our submission.

Sincerely,

Martina Mchenga, PhD-Corresponding Author

Center for Social Science Research (CSSR)

University of Cape town

Rondebosch,

Western Cape

South Africa

Email: martinamchenga@gmail.com

---

## [Decision Letter · Decision Letter 1]

18 Apr 2023

PONE-D-22-22739R1Mental Health is Just an Addendum: Assessing stakeholders’ perceptions on mental health services provision in Malawi prior to and during COVID-19PLOS ONE

Dear Dr. Mchenga,

Thank you for submitting your manuscript to PLOS ONE. After careful consideration, we feel that it has merit but does not fully meet PLOS ONE’s publication criteria as it currently stands. Therefore, we invite you to submit a revised version of the manuscript that addresses the points raised during the review process.

We look forward to receiving your revised manuscript.

Kind regards,

Frank Kyei-Arthur, Ph.D.

Academic Editor

PLOS ONE

Journal Requirements:

**Additional Editor Comments:**

The authors should address the comments/concerns of all the reviewers, paying attention to the following:

1. The abstract

2. The methods (sampling approach, sample size, and selection of participants for study)

3. The results (enhancing the socio-demographic information of participants, and comparing findings on "prior to" and "during COVID-19"

4. Addressing previous comment on why the study did not covered the effectiveness of mental health services?.

Reviewers' comments:

Reviewer's Responses to Questions

**Comments to the Author**

1. If the authors have adequately addressed your comments raised in a previous round of review and you feel that this manuscript is now acceptable for publication, you may indicate that here to bypass the “Comments to the Author” section, enter your conflict of interest statement in the “Confidential to Editor” section, and submit your "Accept" recommendation.

Reviewer #1: (No Response)

Reviewer #3: (No Response)

Reviewer #4: All comments have been addressed

2. Is the manuscript technically sound, and do the data support the conclusions?

Reviewer #1: Partly

Reviewer #3: Yes

Reviewer #4: Yes

3. Has the statistical analysis been performed appropriately and rigorously? 

Reviewer #1: Yes

Reviewer #3: N/A

Reviewer #4: N/A

4. Have the authors made all data underlying the findings in their manuscript fully available?

Reviewer #1: Yes

Reviewer #3: No

Reviewer #4: Yes

5. Is the manuscript presented in an intelligible fashion and written in standard English?

Reviewer #1: Yes

Reviewer #3: Yes

Reviewer #4: Yes

6. Review Comments to the Author

Reviewer #1: 1. Authors have made substantial efforts but the manuscript still needs clarification.

Authors entitled the manuscript as « Mental Health is Just an Addendum: Assessing stakeholders’ perceptions on mental health services provision in Malawi prior to and during COVID-19". When we carefully look at the term “prior to”and “during COVID-19”, I do believe that the two periods should be compared in the manuscript.

2. Regarding the sample size, here is again my concern

-If there are few mental health services in Malawi, it would have been better to interview all the staff members in each health facilities, if not you have to highlight that as study limitation.

-If there are only nine people in those health facilities, I totally agree with you.

Please clarify which one of the above options you stand on?

Reviewer #3: 1. The authors have provided a very good background of the topic and justified why the study fits in Malawi.

2. In my view, the authors have addressed most of the concerns raised by the reviewers. The only point no adequate response is made is the general comment made by reviewer 2. The reviewer expressed concern as to why the study has not covered effectiveness of mental health services. No specific response addressed that particular concern either in the manuscript or in the response letter.

Reviewer #4: This study aimed to assess the stakeholders’ perception on the state of mental health services provision in Malawi and how the COVID-19 pandemic could have impacted the mental health sector. Taking the scarcity of literature on this phenomenon, the authors clearly reviewed the existing literature and brought the rationale for their study into focus. Given the novelty of such studies about the impact of covid-19 on both health systems and individual mental wellbeing, the authors are commended for coming up with such work. I however have a few comments that could improve the manuscript before its published

Abstract:

Methods

When I see a statement like “We purposively sampled nine stakeholders” it automatically appears to me that the sample was pre-determined and yet, the sample size should have been determined at a saturation point.

Conclusion: The authors only state the conclusion from the participants(e.g.,) “The stakeholders also recommended integration of mental health services in all levels of care to improve service quality and efficiency’ Information gathered from the participants should only appear in the result’s section. I wonder why the author’s personal conclusive remarks based on the study’s findings are missing.

Methodology.

Although I am cognizant that saturation can be reached by interviewing few participants especially using phenomenological approaches, the authors should provide detailed information in the methods why their sample size was reached at nine participants and yet there are several other significant stake holders that could have participants in this study. Unless if some professions aren’t considered in the country’s mental health systems structure, I miss to see Psychiatrists/Consultants/Psychologists and Psychiatry Clinical officers and more nurses at different facility levels.

Results.

I suggest that Table1 be moved to results section. Otherwise, it still appears to me that the sample size was pre-determined. Even though the study is qualitative, it would improve their manuscript, if the authors added more information on the participants demographics e.g., age, time spent in service, qualifications, levels of health facilities and any other relevant information.

Discussion

Ok

7. PLOS authors have the option to publish the peer review history of their article (what does this mean?). If published, this will include your full peer review and any attached files.

Reviewer #1: No

Reviewer #3: No

Reviewer #4: **Yes: **Herbert E.Ainamani

While revising your submission, please upload your figure files to the Preflight Analysis and Conversion Engine (PACE) digital diagnostic tool, https://pacev2.apexcovantage.com/. PACE helps ensure that figures meet PLOS requirements. To use PACE, you must first register as a user. Registration is free. Then, login and navigate to the UPLOAD tab, where you will find detailed instructions on how to use the tool. If you encounter any issues or have any questions when using PACE, please email PLOS at figures@plos.org. Please note that Supporting Information files do not need this step.<quillbot-extension-portal></quillbot-extension-portal>

---

## [Author Response · Author response to Decision Letter 1]

20 May 2023

Frank Kyei-Arthur, Ph.D

Academic Editor

PLOS ONE

19/05/2023

Dear Dr. Kyei-Arthur,

Thank you for inviting us to submit a revised draft of our manuscript entitled, " Mental Health is Just an Addendum: Assessing stakeholders’ perceptions on mental health services provision in Malawi prior to and during COVID-19" to PLOS ONE. We also appreciate the time and effort you and each of the reviewers have dedicated to providing insightful feedback on ways to strengthen our paper. Thus, it is with great pleasure that we resubmit our article for further consideration. We have incorporated changes that reflect the detailed suggestions you have graciously provided. We also hope that our edits and the responses we provide below satisfactorily address all the issues and concerns you and the reviewers have noted.

To facilitate your review of our revisions, the following is a point-by-point response to the questions and comments delivered in your letter dated 18 April, 2023.

A. Editor’s comments

1. The abstract

Response- This has been revised

2. The methods (sampling approach, sample size, and selection of participants for study)

Response- We made clarifications on this and responded to the reviewers’ query

3. The results (enhancing the socio-demographic information of participants, and comparing findings on "prior to" and "during COVID-19"

Response- The title has been revised to reflect the content captured in the study. We have also made clarifications on why for this type of study and audience we did not collect data on demographic characteristics of the respondents.

4. Addressing previous comment on why the study did not covered the effectiveness of mental health services?.

Response- Thank you Editor, this has been addressed.

B. Reviewer 4

This study aimed to assess the stakeholders’ perception on the state of mental health services provision in Malawi and how the COVID-19 pandemic could have impacted the mental health sector. Taking the scarcity of literature on this phenomenon, the authors clearly reviewed the existing literature and brought the rationale for their study into focus. Given the novelty of such studies about the impact of covid-19 on both health systems and individual mental wellbeing, the authors are commended for coming up with such work. I however have a few comments that could improve the manuscript before its published 

Abstract: 

1. Methods. When I see a statement like “We purposively sampled nine stakeholders” it automatically appears to me that the sample was pre-determined and yet, the sample size should have been determined at a saturation point. 

Response- in this study, we utilised purposive sampling to differentiate it from random sampling. We deliberately selected individuals with experience in the mental health sector for the interviews, and the number of participants was determined based on data saturation. The decision to use purposive sampling was solely based on the characteristics of our target population, and not on the number of interviews conducted. 

To avoid any confusion, we have rephrased the sentence to provide a clear explanation of the concept of purposive sampling as used in this study. Find the paraphrased sentence below

“We conducted a formative qualitative assessment using key informant interviews (KIIs) among stakeholders with expert knowledge and involvement of mental health services in Malawi. Data were collected till saturation point”.

Also refer to the systematic review here that found that a sample size of between 9-17 interviews reached saturation

2. Conclusion: The authors only state the conclusion from the participants(e.g.,) “The stakeholders also recommended integration of mental health services in all levels of care to improve service quality and efficiency’ Information gathered from the participants should only appear in the result’s section. I wonder why the author’s personal conclusive remarks based on the study’s findings are missing.

Response- Thank you so much for this comment. The conclusion has been revised to reflect our personal conclusive remarks. Find below the rewritten section of the abstract that captures our reflections.

“To address these challenges, policymakers in Malawi could prioritise mental health and integrate mental health services into primary healthcare services to expand access Additionally, policymakers could explore partnerships with the private sector to address shortages of mental health professionals and medication. Overall, this study highlights the need for policymakers to invest in the mental health sector to improve service quality and efficiency, particularly in light of the COVID-19 pandemic”.

3. Methodology. Although I am cognizant that saturation can be reached by interviewing few participants especially using phenomenological approaches, the authors should provide detailed information in the methods why their sample size was reached at nine participants and yet there are several other significant stake holders that could have participants in this study. Unless if some professions aren’t considered in the country’s mental health systems structure, I miss to see Psychiatrists/Consultants/Psychologists and Psychiatry Clinical officers and more nurses at different facility levels. 

Response- Thank you for your valuable feedback on our methodology. We appreciate your concern regarding our sample size of nine participants and the exclusion of certain stakeholder groups from our study. We agree that it is important to provide a detailed explanation of our sample size and stakeholder selection process.

In our study, we aimed to explore the state of mental health services in Malawi, with a specific focus on the challenges and opportunities in the sector. As we mentioned in our paper, the mental health system in Malawi is still developing, and there are only a limited number of facilities and stakeholders available for us to draw from. You can see from this reflective essay that the public facilities have three trained psychiatric nurses and two consultant psychiatrists. For this study we interviewed three psychiatric nurses and one psychiatric/psychologist who owns a private practice. 

We have provided this information in Results- Study population section.

No Position Gender Institution affiliation

KI1 Programme manager Female Private not for profit facility 

KI2 Psychiatric nurse Female Government district hospital

KI3 Psychiatric nurse Female Government tertiary hospital

KI4 Director Male Ministry of Health

KI5 District Health and Social Service Officer Male District hospital

KI6 Private Mental Health Counsellor/Psychiatric consultant Male Private practice

KI7 District Health Officer Female Ministry of Health

KI8 Psychiatric nurse Male Mental health association of Malawi

KI9 Mental hospital director Female Government tertiary hospital

Furthermore, refer to the systematic review here that found that a sample size of between 9-17 interviews reached saturation

4. Results. I suggest that Table1 be moved to results section. Otherwise, it still appears to me that the sample size was pre-determined. Even though the study is qualitative, it would improve their manuscript, if the authors added more information on the participants demographics e.g., age, time spent in service, qualifications, levels of health facilities and any other relevant information. 

Response. Thank you for this comment and suggestion. We included table 1 under study setting and population to provide a description of the people we interviewed. We felt that it made sense that readers have an understanding on the sample before delving into the main results. However, as per your suggestion, we have moved the table to the results section.

We interviewed policy makers and providers of health services- We did not collect information on the age of participants as we did not think it would have an impact on their opinions and perceptions. However, we do agree that including demographic information can enhance the manuscript in certain contexts, but it is not necessary or relevant for the current study's focus. 

Discussion 

Ok

C. Reviewer #1

Authors have made substantial efforts but the manuscript still needs clarification.

1. Authors entitled the manuscript as « Mental Health is Just an Addendum: Assessing stakeholders’ perceptions on mental health services provision in Malawi prior to and during COVID-19". When we carefully look at the term “prior to” and “during COVID-19”, I do believe that the two periods should be compared in the manuscript.

Response- Thank you for this comments and we understand the confusion in the title and have therefore changed it to truly reflect what the study captures. The new title is suggested below

“Mental Health is Just an Addendum: assessing stakeholder’s perceptions on Covid-19 and mental health services provision in Malawi”

2. Regarding the sample size, here is again my concern. If there are few mental health services in Malawi, it would have been better to interview all the staff members in each health facilities, if not you have to highlight that as study limitation. If there are only nine people in those health facilities, I totally agree with you. Please clarify which one of the above options you stand on?

Response- Thank you for this comment. We appreciate your concerns about the sample size. To clarify, we had a specific target audience of respondents for our study, and we stopped the interviews once we reached the saturation point. We did not want to continue the interviews just to increase the number of respondents beyond the point of saturation. Furthermore, the mental health sector is relatively small, with only three trained psychiatric nurses and two consultant psychiatrists see the paper here. Additionally, refer to the systematic review here that found that a sample size of between 9-17 interviews reached saturation. Therefore, we believe our sample is representative of the mental health sector in Malawi. 

Thank you for raising this issue, and we have addressed it in the methodology, results and discussion sections.

D. Reviewer #3

1. The authors have provided a very good background of the topic and justified why the study fits in Malawi.

Response- Thank you so much

2. In my view, the authors have addressed most of the concerns raised by the reviewers. The only point no adequate response is made is the general comment made by reviewer 2. The reviewer expressed concern as to why the study has not covered effectiveness of mental health services. No specific response addressed that particular concern either in the manuscript or in the response letter

Response- Thank you for your comment. We appreciate your acknowledgment that we addressed most of the concerns raised by the reviewers. We also appreciate the specific concern raised by Reviewer 2 regarding the effectiveness of mental health services, which we did not specifically address in the manuscript or response letter

We apologize for not adequately addressing this concern. Our primary focus was to explore the current state of mental health services in Malawi, including their availability, provision, challenges, and effect of the pandemic. However, we agree that it would have been useful to investigate the effectiveness of mental health services as well, and we will take this suggestion into consideration for future studies

In the conclusion section, we recommended that future research should investigate the effectiveness of mental health service delivery, including assessing the perceptions of beneficiaries. We also suggested that the Malawi Facility Assessment Survey should include mental health-related services to evaluate the quality of care provided and the satisfaction of beneficiaries

Thank you again for bringing this concern to our attention, and we will strive to address it in future research.

---

## [Decision Letter · Decision Letter 2]

19 Jun 2023

PONE-D-22-22739R2Mental Health is Just an Addendum: assessing stakeholder’s perceptions on COVID-19 and mental health services provision in MalawiPLOS ONE

Dear Dr. Mchenga,

Thank you for submitting your manuscript to PLOS ONE. After careful consideration, we feel that it has merit but does not fully meet PLOS ONE’s publication criteria as it currently stands. Therefore, we invite you to submit a revised version of the manuscript that addresses the points raised during the review process.

**Mental Health is Just an Addendum: assessing stakeholder’s perceptions on COVID-19 and mental health services provision in Malawi.** This is a good study that focused on the integration of mental health services to primary healthcare and in support of the provision of quality healthcare. However, the organization of the manuscript was poorly done and not well written with greater focus on the Abstracts, Methods, Results, and Conclusion sections. These needs to be promptly addressed.

**Abstract**

“…We conducted a formative qualitative assessment using key informant interviews (KIIs) among stakeholders with expert knowledge and involvement of mental health services in Malawi. Data were collected till saturation point. The stakeholders were classified into four policy circles or networks: national government stakeholders, private for-profit stakeholders, public facility hospital managers and district level officers. We conducted semi-structured interviews and, used thematic analysis to explore the data …”

This is the method section of the abstract and inadequate.

Of what need in the categorization on the stakeholders into 4. What does it add to the study? Also, you only interviewed 9 persons.

Look at the example below, which you can improve upon:

“..We conducted a formative qualitative assessment i.e., Key Informant Interview (KIIs) using a semi-structured interview guide among key stakeholders involved in the delivery of mental health services in Malawi. The interviews were audio recorded in English language and was transcribed for thematic analysis by generating codes re-classified into themes, subthemes, and quotes…….

“…..The findings revealed lack of prioritization and inadequate funding for mental health services, which was exacerbated by the COVID-19. Although mental health services are part of the essential health care package and, therefore are supposed to be provided for free in public facilities at all levels, the services are centralized and only functional at a tertiary level of care in public facilities. Lack of medication, shortage of health workers specialized in mental health and poor governance were also identified as challenges that the pandemic exacerbated…..”

This is the result section of the abstract:

Please, this is a result and not a discussion. This is poorly written. What did you find? Looking at your interview guide you can organise your line of though and aligned your result accordingly and in chronological order.

“…..To address these challenges, policymakers in Malawi should prioritise mental health and integrate mental health services into primary healthcare services to expand access. Additionally, policymakers should explore partnerships with the private sector to address shortages of mental health professionals and medication…..”

This is the conclusion and recommendations:

After the findings what was your conclusion with regards to your title and objectives. What you wrote here is farther from it. Kindly look at your study results and give a concise conclusion with appropriate recommendations.

“…Overall, this study highlights the need for policymakers to invest in the mental health sector to improve service quality and efficiency, particularly in light of the COVID-19 pandemic……. This is not a conclusion as most of this was not features in your study. i.e., service quality and efficiency.

**Keywords**: Please use MeSH in selecting your keywords at least 6

**Introduction**

“…..This paper, therefore, sought to understand from the stakeholders’ perspectives, the state of mental health services provision in a context where resources are limited and how the COVID-19 pandemic affected the provision of the services. We hope that through this study we are engaging with the dialogue on the importance of investing in mental health services in LMICs and strengthen the system for effective provision of the services in the wake of the pandemic and beyond…….” These are the objectives and the justification for the study.

**Methods**

**Study Design**

“This part informed the development of the semi-stricture interview guide.” Please check the spelling of the semi-structure

**Study Settings**

This was grossly inadequate and did not focus on the study setting but rather on data collection method.

I expected the author to give a context of the health service delivery in Malawi with the intersection of mental health service delivery. How is the setup for the health and mental health system in Malawi? How many health facilities, their classifications, and proportion that offer mental health services, what is the spread in the country? What is the administrative structure of Malawi? Total population?

The section gives the audience a general picture of health services and mental health service delivery in Malawi at the time prior to the study. **Study Population and Sampling** Bring it here.

**Data collection tool and method**

Kindly briefly describe some key questions from the interview guide in this section to aid audience’s understanding of your result.

For example: The KII semi-structured interview guide assessed the informant on the awareness of mental health guidelines, policies, and strategies in Malawi, the distribution, availability and access to mental health services, availability of resources for mental health services i.e., human and financial resources, the organization’s role in the provision of mental health services, the impact of COVID-19 on the access and provision of mental health service delivery, and what are the gaps and opportunities experienced in the provision of mental health services resulted from COVID-19 pandemic.

This may give the audience a view of what to expect in your result and it may guide the chronological arrangement of the result.

“Transcripts (of what?) were returned to the participants for comments and data validation.” Please, let this be “Transcripts of the audio recordings were returned to the participants for their comments and corroborative data validation”.

**Data analysis**

There is no where that you mentioned if the analysis was manual or with a software?

**Change “Study population” to “Study population and sampling” (move this up to methods)**

Start by describing the criteria for the population from which your purposely selected the informants. By the time you describe your study setting, it will justify the population selected and if the number is adequate to generate the result. Though a study says 9 – 17 for saturation may not mean your can be 9 which is a minimum. Remember, you segregated your health organizations to four i.e., Public – Federal, Private non-profit, Private for profit, and Public-District health level or secondary level. How many eligible informants operates at this organizational level or health facilities and how many did you selected from these for interview?

Kindly add the sub-section on “Study population and sampling” to the methods section.

**Results**

This section focused on what was found, and no need to discuss the findings until you reached the discussion section. For example, “Types of services by provider” subsection was not written as a result but as a study setting.

The author needs to analyze the informants’ responses as a unit representing Malawi. There is no need differentiating it to the four different health organizations. The respondents’ responses are to be woven into one meaning and supported it with 2 or more quotes.

The author can re-organize this section to clearly follow chronological meaning for the audience.

Looking at your questionnaire guide and using the deductive approach, I expected your results with thematic and sub-thematic headings and quotes that address and arranged chronologically as follows:

• awareness of mental health guidelines, policies, and delivery strategies in Malawi

• the distribution, availability, and access to mental health services

• availability of resources for mental health services i.e., human, and financial resources e.t.c.,

• the organization’s role in the provision of mental health services

• the impact of COVID-19 on the access and provision of mental health service delivery during pandemic

• the gaps and opportunities experienced in the provision of mental health services resulted from COVID-19 pandemic.

So, there may be a need for further analysis and bring out findings that address the above with some supportive quotes.

The author can redo Table 2 by looking at the themes and Sub-themes.

Also, there are some conflicts in your result. Primary level is involved in mental health services because they were given tools like guidelines and triaging tools for identification and referral of cases. But you stated the opposite. Involvement and not implementing was different from if you wrote primary level were not involved. This is what I found in your result “so much that health care providers at community and primary levels of care in public facilities lack skills to effectively provide mental health services of any kind….” I think some skills were given but not been implemented. See this “These include (i) the mental health handbook developed in collaboration with the Scotland Malawi health education partnership. Nearly, all facilities at all levels have the book which specifically targets general practitioners who may have had no training in mental health to act as a guideline. Recently, the guide was made available in soft copy and was launched in 2020” and this “At the primary level, mental health illness assessment skills are taken as first aid; that's why there is no specialized care at that level. The expectation for every health worker at this level, whether a nurse, clinical officer, or medical assistant, is the ability to assess and provide treatment to calm a psychotic/aggressive patient before referring them to a higher facility level (KI 2).” So, these inconsistencies are many in the result section.

Also, on results:

“Inadequate funding and drug shortages

One of the respondents indicated that another big challenge with providing mental health services is inadequate funding. The WHO estimates that for LMICs to effectively provide primary health care services, they need at least $86, however, the current health expenditure per capita in Malawi is $39.90 (28) which is more than 50% less than what is recommended. This means that even fewer resources are allocated to mental health which is not prioritized but falls under non-communicable diseases (NCDs). This inadequate funding also affects medication availability for mental health-related illnesses, especially in public facilities.” The first statement is a result and good to have a quote. However, the following statements are discussions and not what you found in the analysis. These are kept at the discussion section. This approach is clearer and more appropriate than mixing them up.

**Conclusion**

This is not strong and overall conclusion was not well stated by the author. The conclusion should reflect on the title, objectives, and findings from the result to arrive at the overall conclusion from the study found by the authors.

Please, re-write, no conclusion was stated in your conclusion as shown below, but majorly they are recommendations.

Our study engages with the ongoing dialogue on the need for prioritization of mental health services in low- and middle-income countries in the wake of the COVID-19 pandemic. Before any strong recommendations are made, addition studies and analysis are needed in the mental health field to fully unpack the prevalence of mental health disorders in Malawi by sex and age for efficient targeting. We would also need to understand the extend of health system challenges to effectively provide mental health care services for example mapping out available mental health providers, skills, and type of services they can provide. This can be done through the inclusion of mental health related questions in the health facility assessment surveys in Malawi. This will quantify the challenge in effective mental health services delivery and provide a clear understanding of the existing gaps in skills which can then inform appropriate interventions.

Another relevant area of future research would be to understand individual’s perceptions on mental health disorders and how that affect uptake of the services. Studies from Ethiopia (31) and South Africa (32) shows that lack of knowledge and stigma are the main contributing factors that hinder people from seeking mental health services. Finally, given the non-existence of mental health services at lower levels of care, it would also be helpful to understand what interventions are being put in place to promote integration of mental health services at all levels. A report by the WHO and WONCA (20) study shows that integrating mental health services into primary care with effective and acceptable interventions increases access to mental health services.

**These is no information on the limitations to this study. Kindly provide some.**

We look forward to receiving your revised manuscript.

Kind regards,

Obafemi Joseph Babalola, M.D, MPH

Academic Editor

PLOS ONE

---

## [Author Response · Author response to Decision Letter 2]

22 Nov 2023

Reviewer 4

Mental Health is Just an Addendum: assessing stakeholder’s perceptions on COVID-19 and mental health services provision in Malawi.

This is a good study that focused on the integration of mental health services to primary healthcare and in support of the provision of quality healthcare. However, the organization of the manuscript was poorly done and not well written with greater focus on the Abstracts, Methods, Results, and Conclusion sections. These needs to be promptly addressed.

Abstract

“…We conducted a formative qualitative assessment using key informant interviews (KIIs) among stakeholders with expert knowledge and involvement of mental health services in Malawi. Data were collected till saturation point. The stakeholders were classified into four policy circles or networks: national government stakeholders, private for-profit stakeholders, public facility hospital managers and district level officers. We conducted semi-structured interviews and, used thematic analysis to explore the data …”

This is the method section of the abstract and inadequate.

Of what need in the categorization on the stakeholders into 4. What does it add to the study? Also, you only interviewed 9 persons.

Look at the example below, which you can improve upon:

“..We conducted a formative qualitative assessment i.e., Key Informant Interview (KIIs) using a semi-structured interview guide among key stakeholders involved in the delivery of mental health services in Malawi. The interviews were audio recorded in English language and was transcribed for thematic analysis by generating codes re-classified into themes, subthemes, and quotes…….

Response-Thanks for your suggestion and we have revised the section accordingly

“…..The findings revealed lack of prioritization and inadequate funding for mental health services, which was exacerbated by the COVID-19. Although mental health services are part of the essential health care package and, therefore are supposed to be provided for free in public facilities at all levels, the services are centralized and only functional at a tertiary level of care in public facilities. Lack of medication, shortage of health workers specialized in mental health and poor governance were also identified as challenges that the pandemic exacerbated…..”

This is the result section of the abstract:

Please, this is a result and not a discussion. This is poorly written. What did you find? Looking at your interview guide you can organise your line of though and aligned your result accordingly and in chronological order.

Response- The results section has been revised accordingly.

“…..To address these challenges, policymakers in Malawi should prioritise mental health and integrate mental health services into primary healthcare services to expand access. Additionally, policymakers should explore partnerships with the private sector to address shortages of mental health professionals and medication…..”

This is the conclusion and recommendations:

After the findings what was your conclusion with regards to your title and objectives. What you wrote here is farther from it. Kindly look at your study results and give a concise conclusion with appropriate recommendations.

“…Overall, this study highlights the need for policymakers to invest in the mental health sector to improve service quality and efficiency, particularly in light of the COVID-19 pandemic……. This is not a conclusion as most of this was not features in your study. i.e., service quality and efficiency.

Response-Thank you so much for this suggestion. The results section has been revised.

Keywords: Please use MeSH in selecting your keywords at least 6

Response- We have taken time to add more MeSH items

Introduction

“…..This paper, therefore, sought to understand from the stakeholders’ perspectives, the state of mental health services provision in a context where resources are limited and how the COVID-19 pandemic affected the provision of the services. We hope that through this study we are engaging with the dialogue on the importance of investing in mental health services in LMICs and strengthen the system for effective provision of the services in the wake of the pandemic and beyond…….” These are the objectives and the justification for the study.

Response- Correct. 

Methods

Study Design

“This part informed the development of the semi-stricture interview guide.” Please check the spelling of the semi-structure

Response-The sentence has been revised, check the study design section

Study Settings

This was grossly inadequate and did not focus on the study setting but rather on data collection method.

I expected the author to give a context of the health service delivery in Malawi with the intersection of mental health service delivery. How is the setup for the health and mental health system in Malawi? How many health facilities, their classifications, and proportion that offer mental health services, what is the spread in the country? What is the administrative structure of Malawi? Total population?

The section gives the audience a general picture of health services and mental health service delivery in Malawi at the time prior to the study.

Response-Thank you for this useful suggestion. The study setting has been revised

Study Population and Sampling

Bring it here.

Response-This has been done

Data collection tool and method

Kindly briefly describe some key questions from the interview guide in this section to aid audience’s understanding of your result.

For example: The KII semi-structured interview guide assessed the informant on the awareness of mental health guidelines, policies, and strategies in Malawi, the distribution, availability and access to mental health services, availability of resources for mental health services i.e., human and financial resources, the organization’s role in the provision of mental health services, the impact of COVID-19 on the access and provision of mental health service delivery, and what are the gaps and opportunities experienced in the provision of mental health services resulted from COVID-19 pandemic.

This may give the audience a view of what to expect in your result and it may guide the chronological arrangement of the result. “Transcripts (of what?) were returned to the participants for comments and data validation.” Please, let this be “Transcripts of the audio recordings were returned to the participants for their comments and corroborative data validation”.

Response-The section has been revised. And sample questions have been added to the section

Data analysis

There is nowhere that you mentioned if the analysis was manual or with a software?

Response-We have explicitly mentioned that transcription and analysis was done manually

Change “Study population” to “Study population and sampling” (move this up to methods). Start by describing the criteria for the population from which your purposely selected the informants. By the time you describe your study setting, it will justify the population selected and if the number is adequate to generate the result. Though a study says 9 – 17 for saturation may not mean your can be 9 which is a minimum. Remember, you segregated your health organizations to four i.e., Public – Federal, Private non-profit, Private for profit, and Public-District health level or secondary level. How many eligible informants operates at this organizational level or health facilities and how many did you selected from these for interview? Kindly add the sub-section on “Study population and sampling” to the methods section.

Response- Thank you for this suggestion. This has been revised

Results

This section focused on what was found, and no need to discuss the findings until you reached the discussion section. For example, “Types of services by provider” subsection was not written as a result but as a study setting. The author needs to analyze the informants’ responses as a unit representing Malawi. There is no need differentiating it to the four different health organizations. The respondents’ responses are to be woven into one meaning and supported it with 2 or more quotes. The author can re-organize this section to clearly follow chronological meaning for the audience.

Looking at your questionnaire guide and using the deductive approach, I expected your results with thematic and sub-thematic headings and quotes that address and arranged chronologically as follows:

• awareness of mental health guidelines, policies, and delivery strategies in Malawi

• the distribution, availability, and access to mental health services

• availability of resources for mental health services i.e., human, and financial resources e.t.c.,

• the organization’s role in the provision of mental health services

• the impact of COVID-19 on the access and provision of mental health service delivery during pandemic

• the gaps and opportunities experienced in the provision of mental health services resulted from COVID-19 pandemic.

So, there may be a need for further analysis and bring out findings that address the above with some supportive quotes.

The author can redo Table 2 by looking at the themes and Sub-themes.

Also, there are some conflicts in your result. Primary level is involved in mental health services because they were given tools like guidelines and triaging tools for identification and referral of cases. But you stated the opposite. Involvement and not implementing was different from if you wrote primary level were not involved. This is what I found in your result “so much that health care providers at community and primary levels of care in public facilities lack skills to effectively provide mental health services of any kind….” I think some skills were given but not been implemented. See this “These include (i) the mental health handbook developed in collaboration with the Scotland Malawi health education partnership. Nearly, all facilities at all levels have the book which specifically targets general practitioners who may have had no training in mental health to act as a guideline. Recently, the guide was made available in soft copy and was launched in 2020” and this “At the primary level, mental health illness assessment skills are taken as first aid; that's why there is no specialized care at that level. The expectation for every health worker at this level, whether a nurse, clinical officer, or medical assistant, is the ability to assess and provide treatment to calm a psychotic/aggressive patient before referring them to a higher facility level (KI 2).” So, these inconsistencies are many in the result section.

Also, on results: “Inadequate funding and drug shortages

One of the respondents indicated that another big challenge with providing mental health services is inadequate funding. The WHO estimates that for LMICs to effectively provide primary health care services, they need at least $86, however, the current health expenditure per capita in Malawi is $39.90 (28) which is more than 50% less than what is recommended. This means that even fewer resources are allocated to mental health which is not prioritized but falls under non-communicable diseases (NCDs). This inadequate funding also affects medication availability for mental health-related illnesses, especially in public facilities.” The first statement is a result and good to have a quote. However, the following statements are discussions and not what you found in the analysis. These are kept at the discussion section. This approach is clearer and more appropriate than mixing them up.

Response-The whole results section has been revised accordingly.

Conclusion

This is not strong and overall conclusion was not well stated by the author. The conclusion should reflect on the title, objectives, and findings from the result to arrive at the overall conclusion from the study found by the authors.

Please, re-write, no conclusion was stated in your conclusion as shown below, but majorly they are recommendations.

Our study engages with the ongoing dialogue on the need for prioritization of mental health services in low- and middle-income countries in the wake of the COVID-19 pandemic. Before any strong recommendations are made, addition studies and analysis are needed in the mental health field to fully unpack the prevalence of mental health disorders in Malawi by sex and age for efficient targeting. We would also need to understand the extend of health system challenges to effectively provide mental health care services for example mapping out available mental health providers, skills, and type of services they can provide. This can be done through the inclusion of mental health related questions in the health facility assessment surveys in Malawi. This will quantify the challenge in effective mental health services delivery and provide a clear understanding of the existing gaps in skills which can then inform appropriate interventions.

Another relevant area of future research would be to understand individual’s perceptions on mental health disorders and how that affect uptake of the services. Studies from Ethiopia (31) and South Africa (32) shows that lack of knowledge and stigma are the main contributing factors that hinder people from seeking mental health services. Finally, given the non-existence of mental health services at lower levels of care, it would also be helpful to understand what interventions are being put in place to promote integration of mental health services at all levels. A report by the WHO and WONCA (20) study shows that integrating mental health services into primary care with effective and acceptable interventions increases access to mental health services.

Response-Conclusion has been revised following the recommendation

These is no information on the limitations to this study. Kindly provide some.

Response-Limitations of the study have been highlighted in the conclusion

---

## [Editor Report · Decision Letter 3]

19 Jan 2024

PONE-D-22-22739R3Mental Health is Just an Addendum: assessing stakeholder’s perceptions on COVID-19 and mental health services provision in MalawiPLOS ONE

Dear Dr. Mchenga,

Thank you for submitting your manuscript to PLOS ONE. After careful consideration, we feel that it has merit but does not fully meet PLOS ONE’s publication criteria as it currently stands. Therefore, we invite you to submit a revised version of the manuscript that addresses the points raised during the review process.

**ACADEMIC EDITOR:**
**See comments below**

We look forward to receiving your revised manuscript.

Kind regards,

Saiendhra Vasudevan Moodley, MMed

Academic Editor

PLOS ONE

Journal Requirements:

Additional Editor Comments:

Thank you for the efforts to address the concerns that have been raised previously. I am satisfied that you have improved your results section. However, your discussion section needs further work. Please expand on why the issues you identified exist e.g. why is there a lack of integration, lack of human resources for mental health etc and how does it compare to other countries in sub-Saharan Africa. What is the evidence from other countries on how these challenges have been overcome and can these strategies be implemented in Malawi? Please move your limitations and recommendations for future research from your conclusion to discussion section. Your conclusion should consist of a maximum of two paragraphs and focus on your key findings and recommendations. At the moment, it reads more like an extension of your discussion.

---

## [Author Response · Author response to Decision Letter 3]

25 Apr 2024

Editors comments

Thank you for the efforts to address the concerns that have been raised previously. I am satisfied that you have improved your results section. However, your discussion section needs further work. Please expand on why the issues you identified exist e.g. why is there a lack of integration, lack of human resources for mental health etc and how does it compare to other countries in sub-Saharan Africa. What is the evidence from other countries on how these challenges have been overcome and can these strategies be implemented in Malawi? Please move your limitations and recommendations for future research from your conclusion to discussion section. Your conclusion should consist of a maximum of two paragraphs and focus on your key findings and recommendations. At the moment, it reads more like an extension of your discussion.

Response: We truly appreciate this important suggestion. Both the discussion and conclusion sections have been revised as suggested.

About the data, the recordings which is the data used for the analysis were discarded as that was promised to the respondents. The respondents provided consent to include their job titles in the manuscripts but not the names for security purposes. The results section of the manuscript is the data, there is nothing else to be uploaded other than what has been shared in the results and discussion sections.

---

## [Editor Report · Decision Letter 4]

30 May 2024

Mental Health is Just an Addendum: assessing stakeholder’s perceptions on COVID-19 and mental health services provision in Malawi

PONE-D-22-22739R4

Dear Dr. Mchenga,

We’re pleased to inform you that your manuscript has been judged scientifically suitable for publication and will be formally accepted for publication once it meets all outstanding technical requirements.

Kind regards,

Tanya Doherty, PhD

Academic Editor

PLOS ONE
---

## [Editor Report · Acceptance letter]

4 Jun 2024

PONE-D-22-22739R4 

PLOS ONE

Dear Dr. Mchenga, 

I'm pleased to inform you that your manuscript has been deemed suitable for publication in PLOS ONE. Congratulations! Your manuscript is now being handed over to our production team.

Kind regards, 

on behalf of

Professor Tanya Doherty 

Academic Editor

PLOS ONE